

# *Rhodelphis edaphicus* sp. nov.—a new lineage of predatory archaeplastids from agricultural soil

Artem O. Belyaev[1,2], Dmitry G. Zagumyonnyi[1,2], Elena A. Gerasimova[2], German A. Sozonov[2] and Denis V. Tikhonenkov[1,2]

[1] Papanin Institute for Biology of Inland Waters, Borok, Yaroslavl, Russia
[2] AquaBioSafe Laboratory, Tyumen State University, Tyumen, Russia

Corresponding authors
Artem O. Belyaev, abelyaev@ibiw.ru
Denis V. Tikhonenkov,
tikho-denis@yandex.ru

## ABSTRACT

Predatory flagellated protists remain extremely poorly studied but often represent the most important deep-branching lineages of eukaryotic evolution. One of the most enigmatic and poorly studied predatory protist groups is Rhodelphidia. These are heterotrophic flagellates, yet belong to the primarily photosynthetic supergroup Archaeplastida and are related to red algae in particular. Here, we describe a new rhodelphid species and the first known soil representative of this group, *Rhodelphis edaphicus* sp. nov., which was isolated into a clonal culture from agricultural soil. The novel species actively phagocytoses the cells of other protists and bacteria. Using light and electron microscopy, we examined its morphology and identified several novel features, including complex tripartite mastigonemes—not previously reported for this taxon—which link rhodelphids with Cryptista. We expanded a previous 18S rRNA dataset for rhodelphids with environmental sequences and revealed the presence of a soil clade to which the new species belongs. A search of soil metabarcoding data yielded several unknown rhodelphid lineages. Analysis of the distribution of known species and environmental DNA data revealed that rhodelphids inhabit diverse geographic locations worldwide and are found in a variety of habitats, including marine and fresh waters, soils, and, most likely, anaerobic bottom sediments near fumaroles. The diversity of modern rhodelphid habitats, including soil ecosystems, highlights the different environments in which early stages of Archaeplastida evolution may have occurred. The identification and comprehensive study of new lineages of basal archaeplastids provides new insights into the complex evolutionary processes associated with early eukaryotic radiation, the emergence of photosynthesis and plastid evolution that gave rise to the diversification of numerous forms of algae and land plants.

## INTRODUCTION

Protists represent a highly diverse and ecologically important assemblage of soil-dwelling eukaryotes (*Bonkowski, 2004*; *De Ruiter, Neutel & Moore, 1995*; *Geisen et al., 2015b*; *Geisen et al., 2017*). Although still poorly understood, soil protist communities harbor extensive biodiversity, according to the results of high-throughput environmental DNA sequencing

(*Bates et al., 2013*; *Fiore-Donno et al., 2016*; *Geisen et al., 2015a*; *Grossmann et al., 2016*; *Harder et al., 2016*; *Mahé et al., 2017*; *Oliverio et al., 2020*; *Singer et al., 2021*; *Burki, Sandin & Jamy, 2021*). The phylogenetic diversity of soil protists has also been shown to be much greater than previously anticipated (*Bates et al., 2013*; *Geisen et al., 2015b*; *Mahé et al., 2017*). In soils, phagotrophic protists play important ecological roles by regulating microbial populations, influencing nutrient cycling and contributing to soil fertility (*Clarholm, 1985*; *Bonkowski, 2004*; *Crotty et al., 2012*; *Gao et al., 2019*). By grazing on prokaryotic and eukaryotic pathogens, they may enhance plant growth and promote soil health through natural biological control (*Geisen et al., 2018*; *Ren et al., 2023*).

Rhodelphids are one of the most recently discovered and poorly studied protist lineages, represented by unicellular, nonphotosynthetic biflagellates that live in aquatic environments (*Gawryluk et al., 2019*; *Prokina et al., 2023*). Strikingly, phylogenomic analyses revealed that the predatory rhodelphids that feed on other single-celled eukaryotes and bacteria belong to the primarily photosynthetic supergroup Archaeplastida, which unites red algae, green algae, land plants, glaucophytes and picozoans. Rhodelphids are related to red algae and possibly Picozoa (*Gawryluk et al., 2019*; *Schön et al., 2021*) and have gene-rich nuclear genomes, along with a relict nonphotosynthetic primary plastid that lacks a genome. This discovery challenges the traditional views of the origins of Rhodophyta and Archaeplastida evolution as a whole, arguing for a mixotrophic ancestor of red algae and possibly all archaeplastids and suggesting that a mixotrophic nutritional mode, including phagotrophy, persisted well into the evolutionary history of the supergroup (*Gawryluk et al., 2019*). Currently, the eukaryotic phylum Rhodelphidia comprises only three known representatives: *Rhodelphis marinus* Tikhonenkov, Gawryluk, Mylnikov *et* Keeling, 2019; *R. limneticus* Tikhonenkov, Gawryluk, Mylnikov *et* Keeling, 2019; and *R. mylnikovi* Prokina, Tikhonenkov, Lopez-García *et* Moreira, 2023, which were isolated from nearshore marine water with coral sand, South Vietnam; nearshore water with organic debris in freshwater Lake Trubin, Ukraine; and nearshore water with bottom sediments in freshwater pond Étang du Manet, France, respectively (*Gawryluk et al., 2019*; *Prokina et al., 2023*). These are oval and slightly laterally compressed cells with an oblique anterior end and two heterodynamic perpendicularly oriented flagella emerging subapically. *Rhodelphis* cells swim fast at the surface of the substrate and in the water column and quickly consume both bacterial and eukaryotic prey. Thus, rhodelphids are likely to be important predators in microbial communities. Rhodelphids were only described in 2019, and it is likely that this distinct phylum of eukaryotes includes significant ecological, morphological and taxonomic diversity that has yet to be discovered.

In this study, we describe a new rhodelphid species and the first known soil representative of this taxon, *Rhodelphis edaphicus* sp. nov., isolated from agricultural soil. This novel species, which actively feeds on other protists and bacteria, was isolated in clonal culture. Using light and electron microscopy, we examined its morphology and identified several novel features not previously reported for rhodelphids. We analysed the phylogeny of *R. edaphicus* sp. nov. on the basis of 18S rRNA gene sequences and studied the global distribution of *Rhodelphis* in marine and soil habitats. We also discuss the importance of

novel basal archaeplastid lineages in untangling complex evolutionary processes and the role of predatory protists in the functioning of soil microbial communities.

# MATERIAL & METHODS

## Ethic statements

The fieldwork was approved and carried out within the framework of the project of the Ministry of Science and Higher Education of the Russian Federation (agreement no. 075-15-2024-563).

## Culture and sample

A culture of the phagotrophic flagellate Psa-1BS was isolated from soil in Kazakhstan (50°55′09.9″N, 71°25′11.9″E) with planted potatoes (*Solanum tuberosum* L.). The soil sample (one g) was resuspended in 100 ml of spring water (Aqua Minerale, PepsiCo) and enriched with a suspension of *Aeromonas sobria* bacteria (strain ICISC19, Institute for Cellular and Intracellular Symbiosis Collection, Russian Academy of Sciences, Russia), incubated at 22 °C in the dark and examined on the third, sixth, and ninth days of incubation (*Tikhonenkov, Mazei & Embulaeva, 2008*). After isolation *via* a glass micropipette, the Psa-1BS strain was propagated using *Parabodo caudatus* (Dujardin, 1841) Moreira, Lopez-Garcia *et* Vickerman, 2004 (strain BAS-1) as prey, which was grown in spring water (Aqua Minerale, PepsiCo) and fed on *Aeromonas sobria* Popoff and Vron, 1981 bacteria. Strains are currently stored in the collection of the AquaBioSafe Laboratory, University of Tyumen, and in the Live Protozoan Cultures at the Papanin Institute for Biology of Inland Waters, Russian Academy of Sciences. All attempts to cultivate *Rhodelphis edaphicus* sp. nov. without eukaryotic prey on either *Aeromonas sobria* or bacteria from the sample were unsuccessful.

## Light microscopy and video

Observations of live cells were carried out using an AxioScope A1 upright microscope (Carl Zeiss, Jena, Germany) equipped with water immersion 63× objectives, phase contrast and Differential Interference Contrast (DIC) optics. For cell culture handling and preparation for electron microscopy, we employed an Axio Observer 5 inverted microscope (Carl Zeiss, Jena, Germany) with 20× phase contrast lenses. Images and video recordings were obtained using an MC-20 digital camera (Lomo-Microsystems, Saint Petersburg, Russia).

## Scanning electron microscopy (SEM)

For scanning electron microscope (SEM) analysis, cells were pelleted by centrifugation and fixed in 2.5% glutaraldehyde prepared in 0.1 M sodium cacodylate buffer (pH 7.2) for 30 min at room temperature (22 °C). Following fixation, the samples were placed onto polycarbonate plates and subjected to a graded ethanol dehydration series (30%, 50%, 70%, 96%, 100%). The samples were then treated with a 1:1 mixture of ethanol and propylene oxide for 10 min, followed by two washes in 100% propylene oxide. After overnight incubation in pure hexamethyldisiloxane, the specimens were dried, mounted on aluminium stubs, sputter-coated with gold, and examined under a JSM-6510LV scanning electron microscope (JEOL, Tokyo, Japan).

## Transmission electron microscopy (TEM)

For whole-mount transmission electron microscopy (TEM) preparations, drops of cell suspension were applied to Formvar-coated grids and exposed to osmium tetroxide vapours (2%) for 10 min. The grids were rinsed with distilled water, stained with 1% uranyl acetate for 20 min, rinsed again, and shadowed with tungsten oxide ($WO_2$) using a JEE-4X vacuum evaporator (JEOL, Tokyo, Japan). The preparations were examined with a JEM-1011 transmission electron microscope (JEOL, Tokyo, Japan).

## 18S rRNA gene sequencing

Cells were collected from Petri dishes when cultures reached near-maximum density after depletion of prey. The biomass was concentrated by centrifugation ($1,000\times$ g, room temperature) onto $0.8\,\mu$m membranes of Vivaclear minicolumns (Sartorius Stedim Biotech, VK01P042). Genomic DNA was extracted using the Master Pure Complete DNA and RNA Purification Kit (Epicentre, MC85200). The 18S rRNA gene was amplified with universal primers EukA and EukB (*Medlin et al., 1988*) using the EconoTaq PLUS GREEN 2X Master Mix (Lucigen, 30033-1). Polymerase chain reaction (PCR) products were purified with a QIAquick PCR Purification Kit (Qiagen, 433160764) and sequenced by the Sanger method with additional internal primers 18SintF (5′-GGTAATTCCAGCTCCAATAGCGTA-3′) and 18SintR (5′-GTTTCAGCCTTGCGACCATACT-3′). Overlapping reads were assembled into consensus sequences using Geneious R7 7.0.6 software (https://www.geneious.com).

## Phylogenetic analysis and geographical distribution

We updated a previously published dataset (*Prokina et al., 2023*) and increased the representation of the clade of Rhodelphidia by including long and short environmental sequences retrieved from the following soil metabarcoding projects: lowland neotropical forests from Ecuador, Panama and Costa Rica (*Mahé et al., 2017*; NCBI BioProject number PRJNA317860); Canadian boreal forests (*Dai et al., 2021*; NCBI BioProject number PRJNA667813); arable soils of France, Slovenia and Germany and grassland soils of Germany and the UK (*Santos et al., 2020*; NCBI BioProject number PRJNA602420); tropical montane rainforests of Ecuador (*Schulz et al., 2023*; ENA BioProject number PRJEB23549 (ERP105307)); rainforest, jungle rubber, rubber plantations and oil palm plantations of Indonesia (*Schulz et al., 2019*; ENA BioProject number PRJEB23943); and terrestrial habitats from East Antarctica (*Pushkareva et al., 2024*; NCBI BioProject number PRJNA936193).The quality of the sequenced reads in the metabarcoding projects was checked using FastQC v0.11.9 (*Andrews, 2014*). Cutadapt v. 3.5 (*Martin, 2011*) was used for primer sequences removal from the reads. We used the DADA2 pipeline (*Callahan et al., 2016*) for further sequence analysis, including quality filtering, read merging (min overlap = 18 bp), chimera removal, and generation of amplicon sequence variants (ASVs). ASVs with singletons and doubletons were removed to mitigate the potential impact of spurious sequences. To annotate the resulting ASVs, we used a modified PR2 reference sequence database version 5.1.0 (*Guillou et al., 2013*; https://doi.org/10.5281/zenodo.7805244), supplemented with *Rhodelphis* sequences from this study.

The resulting phylogenetic matrix had 96 sequences and 3316 sites. Multiple sequence alignment was conducted with the L-INS-i algorithm implemented in MAFFT v7.490

(*Katoh & Standley, 2013*), following an approach similar to that described by *Zagumyonnyi & Tikhonenkov (2024)*. No trimming was applied to the alignment in our analysis. Phylogenetic reconstructions were carried out using both Bayesian inference and maximum likelihood (ML) methods. Bayesian analyses were performed in MrBayes v5.1.16 (*Ronquist et al., 2012*) under the GTR + GAMMA4 + I substitution model. Four Metropolis-coupled Markov chains were run for 20 million generations, and the initial 50% of sampled trees were discarded as burn-in. Convergence of the runs was confirmed by examining log-likelihood plots and other diagnostics with the sump utility, resulting in an average standard deviation of split frequencies of 0.0048. ML trees were generated with IQ-TREE v1.6.12 (*Nguyen et al., 2015*) using 1,000 nonparametric bootstrap replicates. ModelFinder selected the TN+F+R5 model as the best fit for the data.

The Ocean Barcode Atlas (*Vernette et al., 2021*; http://oba.mio.osupytheas.fr) was used to check the geographical distribution of rhodelphids in the World Ocean with implementation of the 'Tara Oceans DADA2 ASVs 18S V4 (eukaryotes)' database and '*Rhodelphis marinus*' as a taxonomic query.

### Electronic publication and life science identifiers

The electronic version of this article in Portable Document Format (PDF) will represent a published work according to the International Commission on Zoological Nomenclature (ICZN), and hence the new names contained in the electronic version are effectively published under that Code from the electronic edition alone. This published work and the nomenclatural acts it contains have been registered in ZooBank, the online registration system for the ICZN. The ZooBank LSIDs (Life Science Identifiers) can be resolved and the associated information viewed through any standard web browser by appending the LSID to the prefix http://zoobank.org/. The LSID for this publication is: urn:lsid:zoobank.org:pub:95653C6F-EDF2-4E27-8552-B08EE46074BA. The online version of this work is archived and available from the following digital repositories: PeerJ, PubMed Central SCIE and CLOCKSS.

## RESULTS

### External morphology and behavioural features of *Rhodelphis edaphicus* sp. nov.

The cells are 10.2–17.7 µm in length ($n = 30$; mean 13.6 µm; median 13.1 µm) and 5.5–8.4 µm in width ($n = 30$; mean 6.8 µm; median 6.8 µm). Typically, a cell has an ellipsoidal shape with an oblique anterior end (Figs. 1A–1C, 1J, 2B, 2F), although some cells can have a cone-shaped form (Figs. 1K, 1L, 2C), and they are usually larger in size. The shape of the cell can also change depending on nutritional (starving/well-fed) conditions or before dividing (Figs. 1E, 1H, 1I, 2A). Two heterodynamic flagella emerge subapically on the right side of the anterior end of the cell from the two flagellar pockets, which are separated by a keel-shaped protrusion that continues as a keel along the ventral side of the cell (Figs. 2A, 2B, 2F inset). The flagella do not have acronemes. On both flagella, we observed complex tripartite mastigonemes and some thin hairs located between the mastigonemes (Figs. 3A–3H). These hairs extended from the flagella (Figs. 3D, 3G, 3H) and

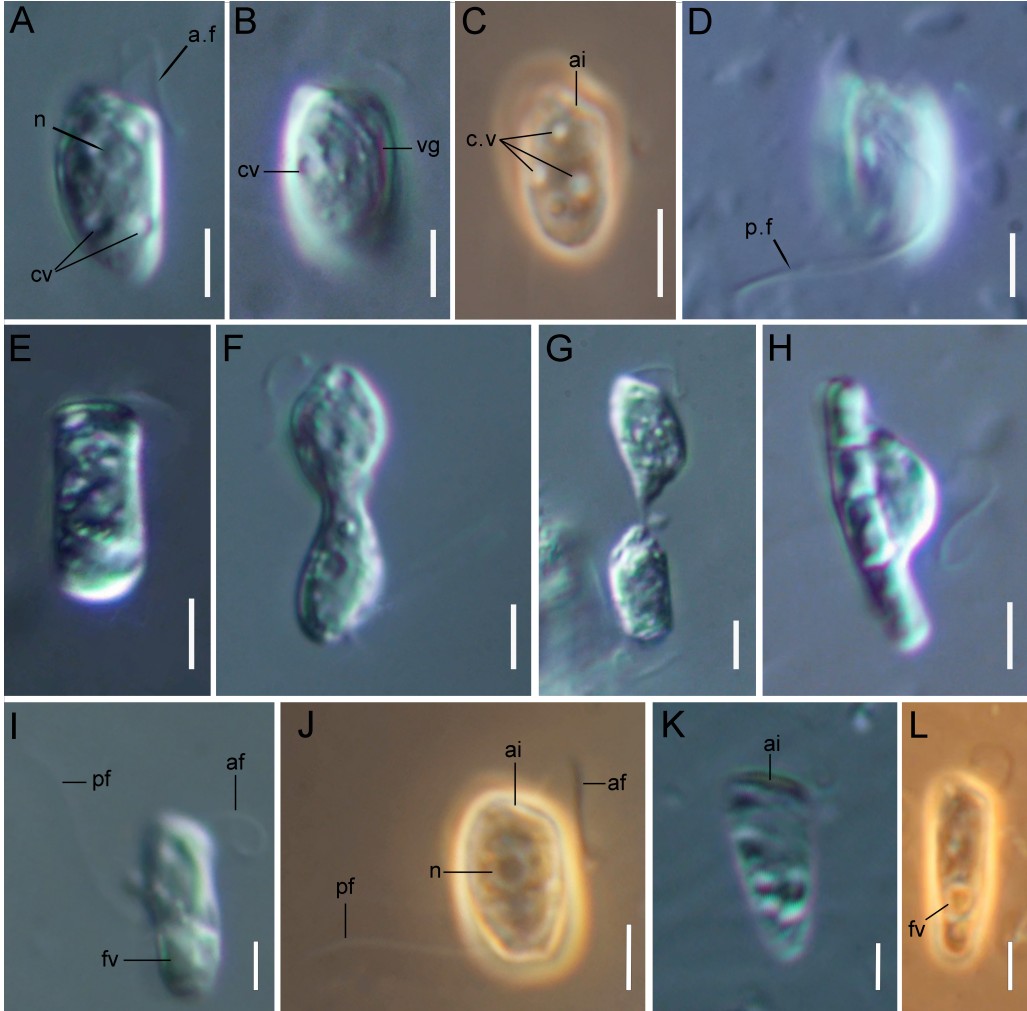

**Figure 1** **Light microscopy of *Rhodelphis edaphicus* sp. nov.** (A–D, J) typical cell shape; (E) cell prior to division; (F, G) dividing cells; (H) *R. edaphicus* sp. nov. ingesting a bacterial colony; (I) well-fed cell; (K, L) cone-shaped large cells. Abbreviations: ai, anterior invagination at the apical end of the cell; af, anterior flagellum; cv, contractile vacuole; fv, food vacuole; n, nucleus; pf, posterior flagellum; vg, ventral groove (A, B, D, E–I, K) DIC contrast; (C, J, L) phase contrast. Scale bar: five μm.

lacked the basal segment (p1) characteristic of tripartite mastigonemes (Figs. 3C, 3F, 3G). They are thinner than the middle segment (p2) of tripartite mastigonemes and resemble the distal part (p3) of complex mastigonemes.

In addition, some mastigonemes appeared to consist of only two parts (the basal and middle segments, p1 and p2), lacking the distal filament (Figs. 3D, 3H), which may be explained by a fixation artifact. Some mastigonemes were found to be much longer than others, possibly due to gaps in the middle part (arrow in Fig. 3G). Mastigonemes were often found detached from the flagella, suggesting that they may have been partially destroyed during fixation. The anterior flagellum, measuring 10.7 to 20.4 μm ($n = 27$; mean 16.3 μm; median 16.9 μm), is directed anteriorly and laterally and exhibits wave-like movement.

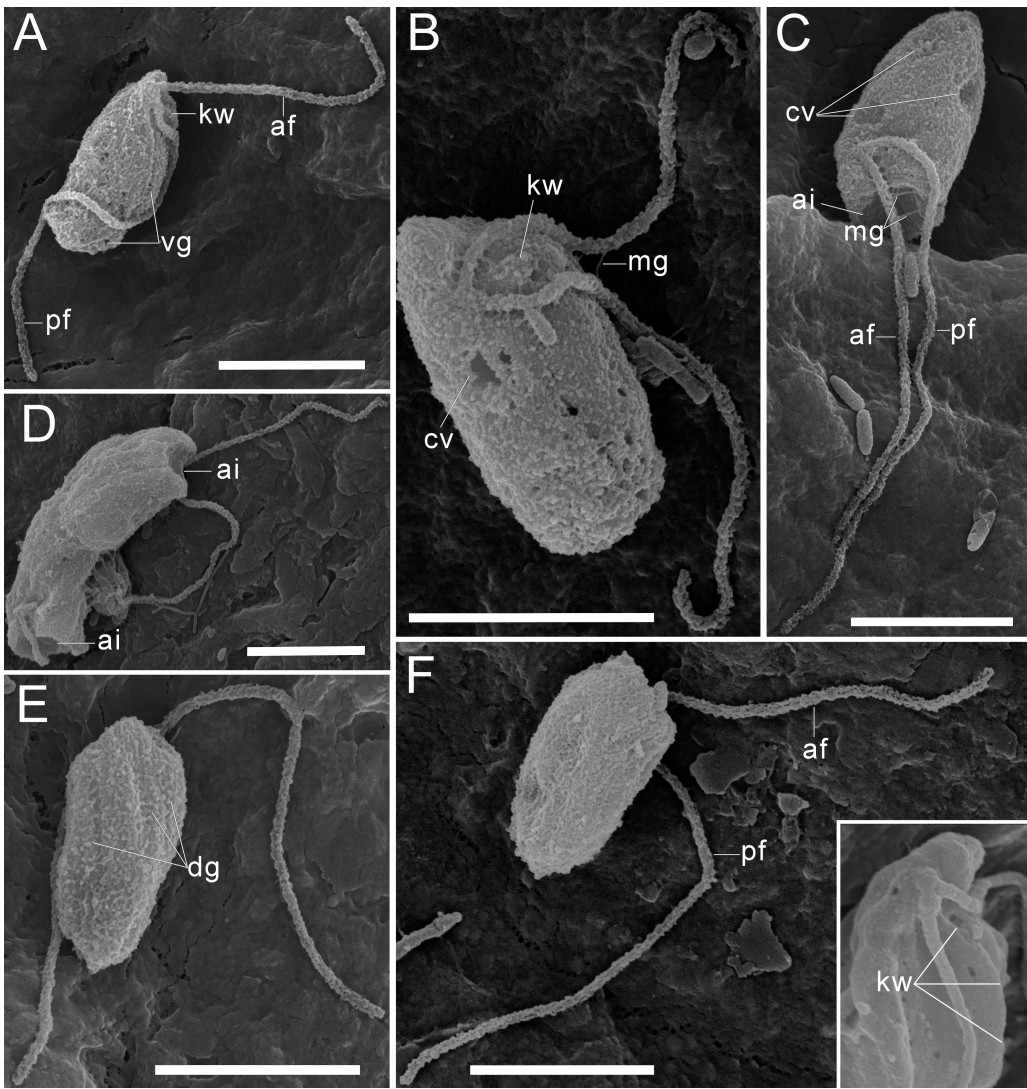

**Figure 2** **External morphology of *Rhodelphis edaphicus* sp. nov., SEM.** (A–B), (F) General view of *R. edaphicus* sp. nov., (C) three contractile vacuole invaginations in *R. edaphicus* sp. nov., (D) dividing cells with anterior invaginations, (E) dorsal view of the cell, (F) (inset)–subapical end of the cell. Abbreviations: ai, anterior invagination at the apical end of predator cell; af, anterior flagellum; cv, contractile vacuole; dg, dorsal grooves on the cell surface; kw, keel-shaped wall between flagellar pockets; mg, mastigonemes; pf, posterior flagellum; vg, ventral groove. Scale bar: five µm.

The posterior flagellum ranges from 18.9 to 34 µm ($n = 25$; mean 26.0 µm; median 25.2 µm) in length and runs along the ventral side of the cell body (Fig. 2A). There is a wide apical invagination resembling a cytostome at the anterior end of the cell, although during feeding, this area probably performs a receptor function in recognizing the prey cell (Figs. 1C, 1J, 1K; 2C, 2D; Video S1). At the beginning of feeding, the predator usually attaches to the prey through the apical invagination and then turns towards it with its posterior end, where phagocytosis of the prey occurs (Video S1), resulting in the complete engulfment of eukaryotic or prokaryotic prey with the formation of a food vacuole. Attachment to the

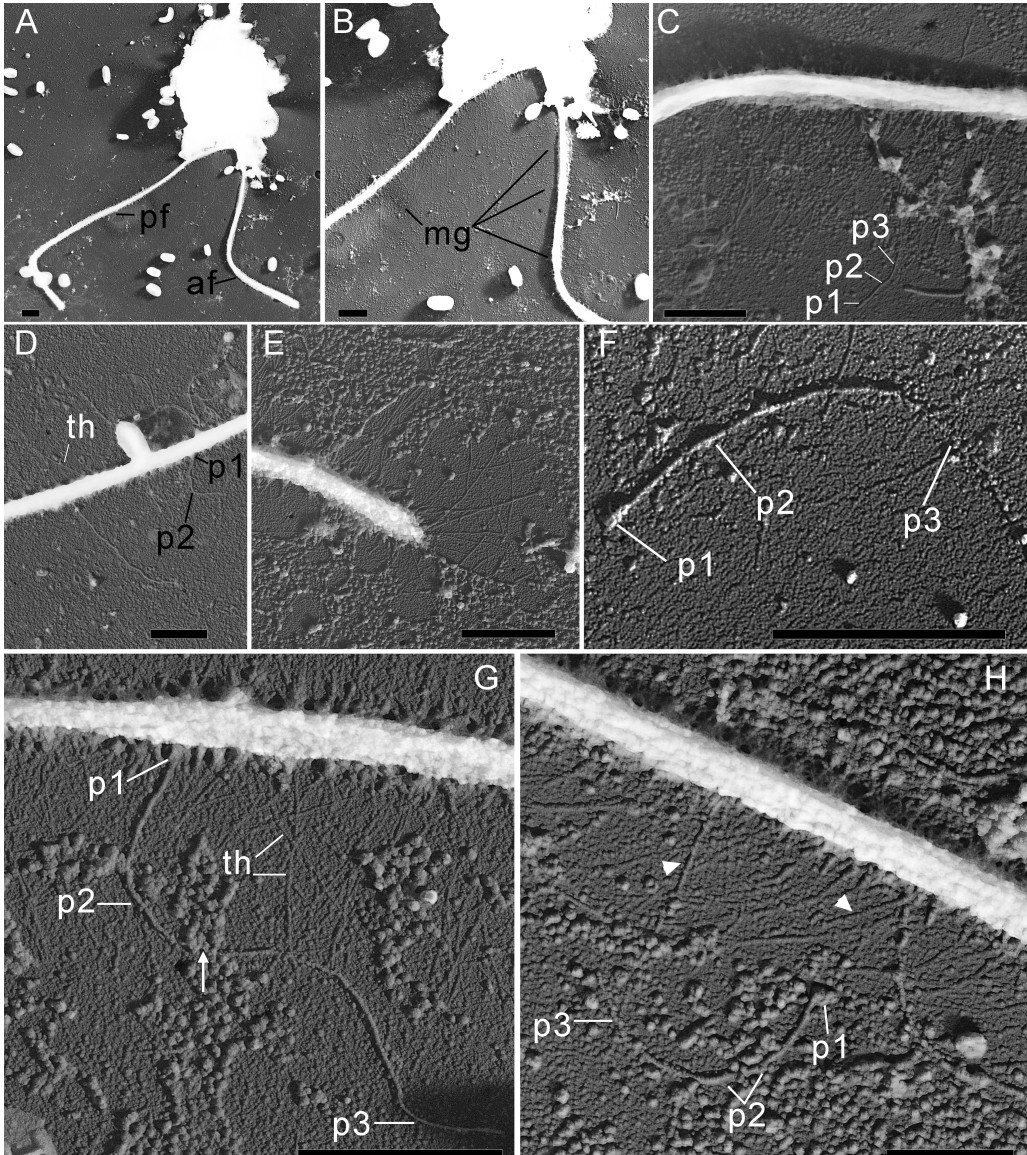

**Figure 3** **The structure of mastigonemes in *Rhodelphis edaphicus* sp. nov., TEM.** (A) General view of the cell, (B) mastigonemes on both flagella, (C) simple and tripartite mastigonemes on the posterior flagellum, (D) simple and bipartite mastigonemes on the anterior flagellum, (E) simple hair s on the anterior flagellum, (F) three parts of the tripartite mastigoneme. Abbreviations: af, anterior flagellum; mg, mastigonemes; p1, p2, p3, parts of mastigonemes; pf, posterior flagellum; th, thin filament hair. Scale bar: one μm.

prey cell in the strain we studied occurs more often than its engulfment. Quite frequently, the floating cells of a predator only attach to the prey by the anterior end, after which they detach and continue to float. Alternatively, the predator may turn towards the prey with its ventroposterior end, where the feeding groove is located (Figs. 1B; 2A), and phagocytosis may occur. The engulfment of the prey cell occurs very quickly, in just 5–10 s (Video S1). During engulfment of larger prey, the cell can stretch, taking on an irregular shape.

In particular, this is observed when a predator engulfs part of a large bacterial colony using the ventral groove (Fig. 1H; Video S2). *Rhodelphis* actively rotates and beats its flagella to detach bacteria from the colony. In our observations, *Rhodelphis* did not survive in culture without eukaryotic prey.

Three contractile vacuoles are usually distinguishable in a cell. They form an almost isosceles triangle with the base closer to the posterior end of the cell and the apex near the anterior end, sometimes shifting dorsally (Figs. 1A–1C; 2B, 2C). SEM revealed that the dorsal surface of the cell bears approximately three longitudinal grooves extending along its entire length (Fig. 2E), although these grooves were not observed under light microscopy and may be an artefact. The nucleus is positioned in the anterior third of the cell, closer to the center of the vertical axis of the cell (Figs. 1A, 1J). Cell division is longitudinal (Figs. 1F, 1G). Cysts were not observed.

### 18S rRNA phylogeny of *R. edaphicus* sp. nov.

We expanded a previous 18S rRNA dataset for rhodelphids (*Prokina et al., 2023*) with environmental sequences and the 18S rRNA gene sequence of the studied soil strain. *R. edaphicus* sp. nov. forms a fully supported clade with soil environmental sequences from Germany and Slovenia according to the results of Bayesian analysis and IQ-TREE reconstruction (Figs. 4, 5), confirming the establishment of a new species associated with the soil habitat. According to the Basic Local Alignment Search Tool (BLAST) results, the 18S rRNA sequence similarity among rhodelphids ranged from 89.9% to 95.2% (Table S1). The 18S rRNA sequence of *R. edaphicus* sp. nov. is only 92.7% identical to that of the closest described relative, freshwater *R. mylnikovi*. Together with the other freshwater strain, *R. limneticus* and environmental sequences from the soil samples in Costa Rica and Indonesia, *R. mylnikovi* forms a clade (Bayesian inference (BPP)=0.98, bootstrap support (ML, %) = 67) sister to *R. edaphicus* sp. nov. grouping (BPP = 0.98, ML = 63). The only marine species known to date, *R. marinus*, forms a basal lineage within the described rhodelphids (BPP = 1, ML = 95). The clade of environmental marine sequences was recovered as a sister (Fig. 5) or in unresolved trichotomy (Fig. 4) with soil and freshwater sequences and isolates, albeit without support. An environmental sequence from anoxic sediment near active fumaroles on a submarine caldera floor at a depth of 200 m (AB191436) occupied a separate and unresolved position within rhodelphids in the 18S rRNA gene phylogeny (Figs. 4, 5). Bayesian analysis and IQ-TREE reconstruction provided moderate to full support for the monophyletic relationship between Rhodelphidia and Rhodophyta (Figs. 4, 5).

### Global distribution of Rhodelphidia according to marine metabarcoding and soil metabarcoding data

A search of soil metabarcoding data yielded four different ASVs related to *Rhodelphis*, which originated from various geographic zones and environmental habitats (Fig. 6), including agro-landscapes of Slovenia (Moskanjci, 18 occurrences) and Germany (Scheyern, 4 occurrences), neotropical forests of Costa Rica (509 occurrences), and tropical lowland rubber jungles in Indonesia (32 occurrences). Across all terrestrial habitats where

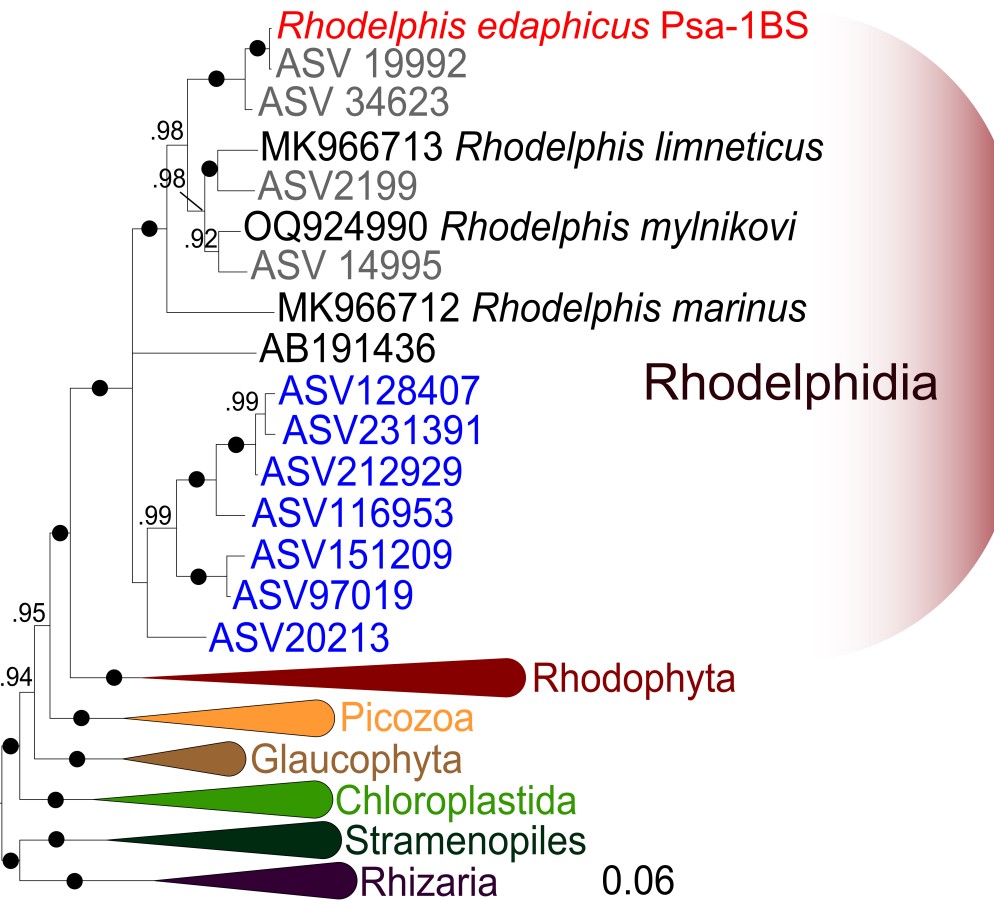

**Figure 4  Bayesian 18S rRNA phylogeny of rhodelphids.** Phylogeny inferred with MrBayes, with branch nodes showing MrBayes posterior probabilities. The black dots indicate nodes with full support. The sequence labels in blue font represent marine organisms sourced from the Tara Ocean Atlas, whereas the labels in gray font represent soil organisms from metabarcoding surveys.

rhodelphids were detected, their proportion relative to all reads ranged from 0.003% to 0.488%.

Multiple ASVs of *Rhodelphis marinus* were found in the Ocean Barcode Atlas dataset, which originated from various regions, including the Mediterranean Sea off the coast of Spain, the South Atlantic near southern Africa, the Indian Ocean, including the Arabian Sea, the South Pacific near Chile, and seas of the Arctic Ocean, including the Kara Sea, East Siberian Sea and Chukchi Sea (Fig. 6). Thus, rhodelphids are distributed worldwide across both hemispheres, including the waters of polar regions and the tropics.

## DISCUSSION

The morphology of *R. edaphicus* sp. nov. includes unique features that clearly distinguish it from other known rhodelphid species. The posterior flagellum of other *Rhodelphis* species possesses thin simple hairs, whereas both flagella of *R. edaphicus* sp. nov. also

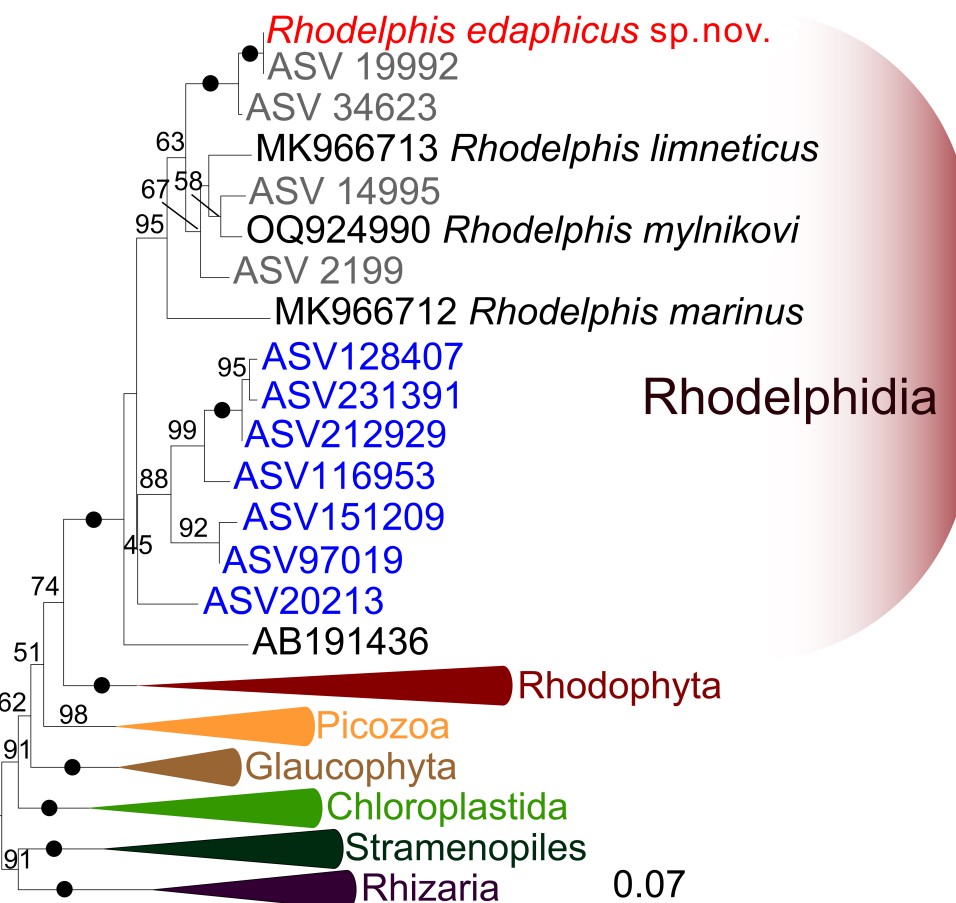

**Figure 5** **Maximum likelihood 18S rRNA phylogeny of rhodelphids.** Phylogenetic tree reconstructed using IQ-TREE, with branch nodes showing IQ-TREE standard bootstrap values. The black dots indicate nodes with full support. The sequence labels in blue font represent marine organisms sourced from the Tara Ocean Atlas, whereas the labels in gray font represent soil organisms from metabarcoding surveys.

possess tripartite mastigonemes. Tripartite mastigonemes of *R. edaphicus* sp. nov. consist of a proximal segment, a medial part, and a terminal filament, similar to the structure described by Kugrens and colleagues for certain cryptomonads (see Fig. 13 in *Kugrens, Lee & Andersen, 1987*). There are many structural variations in the flagella mastigonemes of cryptomonads (*Kugrens, Lee & Andersen, 1987*), and we were unable to determine which type is most similar to those of *R. edaphicus* sp. nov. However, it is clear that *R. edaphicus* sp. nov. possesses a complex of compound mastigonemes on each flagellum, consisting of three distinct segments. The simple hairs on both flagella resemble the distal part of the mastigoneme. Revealing complex mastigonemes in rhodelphids is of critical importance, considering the close evolutionary relationship between Cryptista and Archaeplastida (*Burki et al., 2016*; *Strassert et al., 2019*). This observation sparks the idea that the common ancestor of Cryptista and Archaeplastida may have possessed complex tripartite mastigonemes. Furthermore, the presence of tripartite mastigonemes in Stramenopiles and Telonemia may indicate that this feature is a plesiomorphic trait

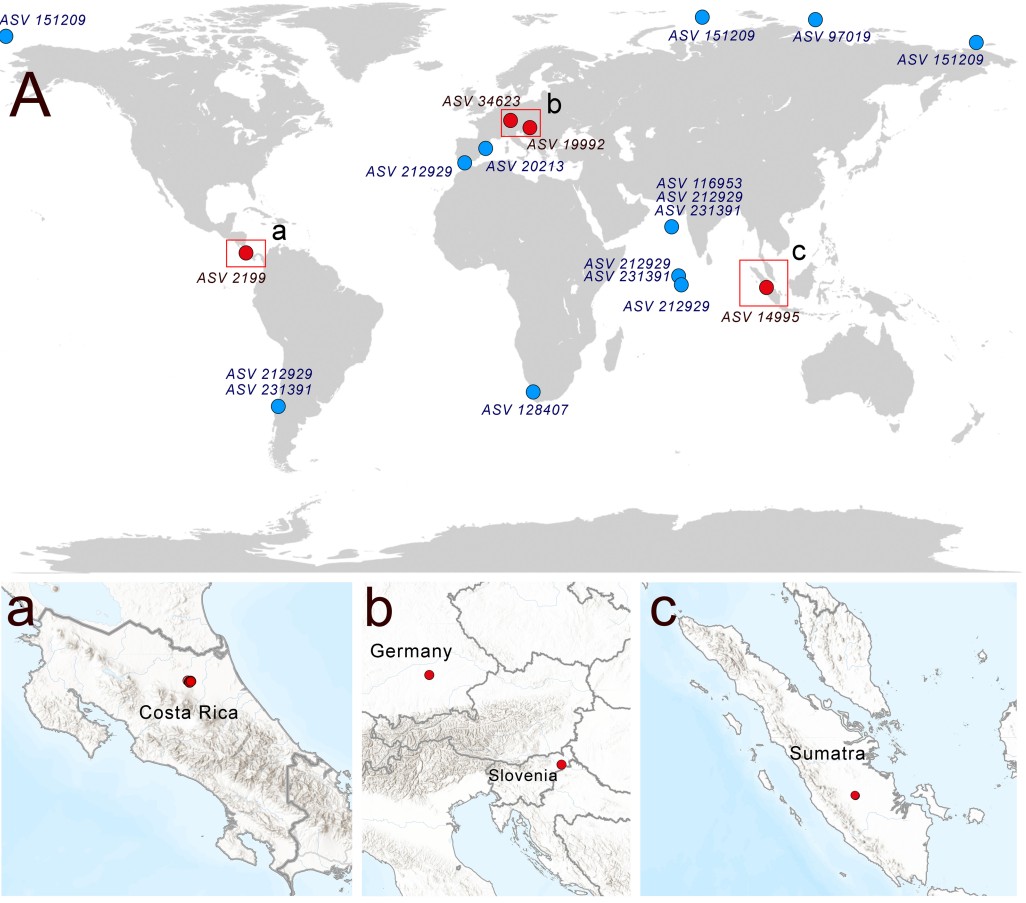

**Figure 6  Distribution of rhodelphids in the World Ocean (blue dots) and soil habitats (red dots).** (A) The general map, (a–c) locations of the soil habitats.

for Diaphoretickes at least (*Shalchian-Tabrizi et al., 2006*; *Tikhonenkov et al., 2022*). Conversely, it cannot be ruled out that complex mastigonemes may evolve independently multiple times.

Another feature that distinguishes *R. edaphicus* sp. nov. from other known rhodelphids is the wide morphological variety of cell shape. The dorsal surface of the cell of *R. edaphicus* sp. nov. has a grooved structure, which is also characteristic of *R. mylnikovi* (*Prokina et al., 2023*) but has not been described for other rhodelphids. In our observations, *R. edaphicus* sp. nov. does not produce pseudopodia as *R. mylnikovi* does and does not exhibit cannibalistic behavior (*Prokina et al., 2023*). At the same time, the feeding process, with initial attachment to the prey by the apical part of the cell and subsequent engulfment of the prey at the posterior cell end, is similar in all known rhodelphid species (*Gawryluk et al., 2019*; *Prokina et al., 2023*). We believe that the apical invagination found in *R. edaphicus* sp. nov. may act as a receptor structure, which is also likely present in other rhodelphids but is less pronounced and has not previously been noted in electron microscopy preparations.

The 18S rRNA gene sequences of *Rhodelphis* species show unusually high divergence within the genus, despite the typically conserved nature of this gene in most eukaryotic lineages. This may simply indicate lineage-specific differences in molecular evolution rates, as divergence in rRNA genes does not necessarily correlate with major biological or ecological differences. These observations illustrate the limited resolution of 18S rRNA data for reconstructing relationships within rhodelphids and emphasize the need for phylogenomic and morphological studies.

It is obvious that rhodelphids inhabit completely different habitats, including marine and fresh waters and soils. The marine environmental sequence AB191436 from anoxic sediment near active fumaroles (*Takishita et al., 2005*), identified as a separate lineage of Rhodelphidia, is extremely interesting. Investigating the morphology and cell biology of such organisms is particularly intriguing, as these features may reflect deep ancestral traits within the group. If so, their anaerobic and thermophilic nature could provide insights into the evolution of metabolism in rhodelphids. The isolation and study of the protist from an anoxic habitat, represented by the above-mentioned environmental sequence, can yield unexpected results in clarifying the origin and early stages of the evolution of archaeplastids. These are important questions since archaeplastids play a key role in terrestrial and aquatic ecosystems, and the multiple and independent enslavements of their cells with primary plastids by cells of other eukaryotes led to the emergence of organisms with secondary and tertiary plastids (*Keeling, 2010*; *Keeling, 2013*; *Miyagishima, Nakanishi & Kabeya, 2011*). This process has played a major role in eukaryotic evolution and eukaryotic diversification and promoted increasing species diversity on the planet. The integration of a photosynthetic cyanobacterium into a phagotrophic protist spurred the radiation of Archaeplastida, and the ancestors of Archaeplastida may have resembled predatory *Rhodelphis* cells morphologically. The cyanobacterial ancestry of primary plastids is no longer debated, and it was found that *Gloeomargarita lithophora*, a cyanobacterium from the microbiolites of alkaline lakes in Mexico, shares the most recent common ancestor with plastids of red, green, and glaucophyte algae, with an inferred freshwater ancestral habitat (*Lewis, 2017*; *Ponce-Toledo et al., 2017*; *Sánchez-Baracaldo et al., 2017*). At the same time, it is possible that the primary plastids of Archaeplastida have multiple origins (*Stiller, 2007*), and the ancestral form of green algae (marine or freshwater) is uncertain (*Leliaert et al., 2016*). Representatives of unicellular archaeplastids include a wide variety of marine species, but at the same time, there is a growing understanding of the prevalence and diversity of freshwater and terrestrial taxa (*Lewis & Lewis, 2005*; *Lewis, 2017*; *Delwiche & Cooper, 2015*). The diversity of habitats, including soil, of modern rhodelphids suggests different options for environments where the early stages of archaeplastid evolution could have occurred.

The discovery of *Rhodelphis edaphicus* sp. nov., the first soil-dwelling representative of Rhodelphidia, expanded our understanding of the ecological diversity of this enigmatic group and demonstrated its presence in terrestrial ecosystems. *R. edaphicus* is a predatory protist that actively consumes both eukaryotic and prokaryotic cells but cannot survive on bacteria alone, indicating its role as a eukaryovorous predator occupying high trophic levels in the soil microbial food web. This highlights the potential ecological importance

of rhodelphids as regulators of microbial communities and participants in nutrient and energy transfer in soils. The broad prey preferences observed in *R. edaphicus* and related lineages suggest that such protists could play underappreciated roles in shaping soil microbial dynamics and plant–microbe interactions, warranting further investigation. Detailed morphological, phylogenomic, and ecological studies of rhodelphids from diverse environments will help clarify their evolutionary history, ecological functions and potential practical applications in microbiome management.

### Taxonomic summary

Assignment. Eukaryota; Archaeplastida; Rhodelphidia; Rhodelphea; Rhodelphida; Rhodelphidae; *Rhodelphis*.

*Rhodelphis edaphicus* sp. nov. Belyaev, Zagumyonnyi, Gerasimova, Sozonov et Tikhonenkov.

Diagnosis. Cells are 10.2–17.7 µm in length and 5.5–8.4 µm in width, with an ellipsoid shape and an oblique anterior end. An anterior invagination is present at the apical end. Two heterokont flagella are present, measuring 10.7–20.4 µm (anterior) and 18.9–34 µm (posterior). A keel-shaped wall separates the flagellar pockets subapically and continues along the ventral surface of the cell. Division is longitudinal. No cysts are observed.

Type locality. Agricultural soil from the southern region of the Astana suburbs (50°55′09.9″N, 71°25′11.9″E).

Type material: The name-bearing type (hapantotype) is the SEM stub Psa-1BS, bearing *Rhodelphis* cells (prey and prokaryotes excluded), deposited in the permanent protist collection of the Papanin Institute for Biology of Inland Waters RAS (Borok, Russia), which maintains institutional access and long-term preservation.

Etymology. Named after its soil habitat.

Gene sequence. The 18S rRNA gene sequence has the GenBank Accession Number PV254715.

Publication ZooBank LSID. urn:lsid:zoobank.org:pub:95653C6F-EDF2-4E27-8552-B08EE46074BA.

Taxon ZooBank LSID. urn:lsid:zoobank.org:act:3D7FAF47-30B3-4A4D-A426-05EC7361C441.

### Funding

This study was supported by the Ministry of Science and Higher Education of the Russian Federation (agreement no. 075-15-2024-563). The funders had no role in study design, data collection and analysis, decision to publish, or preparation of the manuscript.

### Grant Disclosures

The following grant information was disclosed by the authors:
Ministry of Science and Higher Education of the Russian Federation: 075-15-2024-563.

## Competing Interests

The authors declare no competing interests.

## Author Contributions

- Artem O. Belyaev conceived and designed the experiments, performed the experiments, analyzed the data, prepared figures and/or tables, authored or reviewed drafts of the article, and approved the final draft.
- Dmitry G. Zagumyonnyi conceived and designed the experiments, performed the experiments, analyzed the data, prepared figures and/or tables, authored or reviewed drafts of the article, and approved the final draft.
- Elena A. Gerasimova conceived and designed the experiments, analyzed the data, authored or reviewed drafts of the article, and approved the final draft.
- German A. Sozonov performed the experiments, authored or reviewed drafts of the article, and approved the final draft.
- Denis V. Tikhonenkov conceived and designed the experiments, analyzed the data, prepared figures and/or tables, authored or reviewed drafts of the article, and approved the final draft.

## Field Study Permissions

The following information was supplied relating to field study approvals (i.e., approving body and any reference numbers):

Field experiments were approved by the Ministry of Science and Higher Education of the Russian Federation (project number: 075-15-2024-563 "Emergent biological threats to agriculture in Russia and the Commonwealth of Independent States countries in the context of global changes").

## DNA Deposition

The following information was supplied regarding the deposition of DNA sequences:

PV254715.

## Data Availability

The raw data from the phylogenetic analysis, including the nucleotide alignment and phylogenetic trees, video files and other supplementary materials, are available at figshare:

- Belyaev, Artem; Tikhonenkov, Denis (2025). *Rhodelphis edaphicus* sp. nov.—A New Lineage of Predatory Archaeplastids from Agricultural Soil (Supplementary Materials). figshare. Dataset. https://doi.org/10.6084/m9.figshare.28540115.v2.

## New Species Registration

The following information was supplied regarding the registration of a newly described species:

Publication LSID: urn:lsid:zoobank.org:pub:95653C6F-EDF2-4E27-8552-B08EE46074BA

*Rhodelphis edaphicus* LSID: urn:lsid:zoobank.org:act:3D7FAF47-30B3-4A4D-A426-05EC7361C441.

Genus: *Rhodelphis* Tikhonenkov, Gawryluk, Mylnikov & Keeling in Gawryluk, Tikhonenkov, Hehenberger, Husnik, Mylnikov & Keeling, 2019.

## Supplemental Information

Supplemental information for this article can be found online at http://dx.doi.org/10.7717/peerj.20071#supplemental-information.

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
