# Peer review of "Rhodelphis edaphicus sp. nov.—a new lineage of predatory archaeplastids from agricultural soil"

_PeerJ, doi:10.7717/peerj.20071_

## Round 0.1 · original submission · Major Revisions

· Academic Editor

Major Revisions

Reviewer 1 ·

Basic reporting

The manuscript written by Belyaev et al. reports a new species of Rhodelphis, which is a novel and evolutionary important protist group. This new species is phylogenetically distinct from the other Rhodelphis species and probably has different ecological roles in nature. I agreed with the discussion and conclusion in this work, and the findings are highly worth reporting. However, several concerns also remain and I would like to request the authors to address and/or confirm those points before the recommendation for the publication.

Major concerns:
The explanation about the TEM observation is entirely missing in the Results section. The authors must revise it.

I am keen to know the natural state of mastigonemes. The mastigonemes of R. edaphicus seem very thin and fragile. As the results shown in Figure 3 are not so clear, I could not be convinced by the structures. Please show its bi- and tri-partite structures more clearly. I guess that bipartite mastigonemes may be an artifact that is made from tripartite ones during the fixation process. Please mention and discuss it, if the authors agree or disagree with my view. I could not understand ‘thin filament’ either. Is it different from p3? Where does it exist?

R. edaphicus was cultivated from the soil sample, and the cyst formation could not be found under the conditions applied in this study. However, the authors only showed R. edaphicus cells that grew in the liquid medium. How did the authors confirm whether R. edaphicus is active in the soil or not?

Minor points:
Line 200:
In my opinion, ‘wall’ is not appropriate here. Is this decorating the cell from the outside? If not, it may be better to use ‘swelling’, ‘projection’, or something like that.

Please add the information about the size of the 18S rRNA dataset. How many sites were there in the analyzed dataset?

Line 285-286:
I could not understand which data supported this sentence and please explain how it is low.

Experimental design

-

Validity of the findings

-

·

Basic reporting

The paper is a simple new description of a very interesting protist species of rhodelphid. It is engaging and thorough in every section.

Experimental design

All methods/experiments were reported well. Figures are good.

Validity of the findings

I have no comments. Everything is fantastic.

Additional comments

I find the similarity of this species to cryptists amazing. Great paper!

Reviewer 3 ·

Basic reporting

-The language used was clear
-For literature, sometimes more recent papers should be cited, especially when authors insist that the reseacrh described is recent: "according to the results of recent high-throughput environmental DNA (Lines 57-59)." the most recent paper dates from 8 years ago...
-The rest is okay

Experimental design

The methodology used is very appropriate for the experiments described here. It is well described, and research aims are well defined and do fill a scientific gap.

Validity of the findings

Findings are valid and very interesting. Data are robust and available. I have some comments about some parts of the text that do not match the topic (please see the next paragraph).

Additional comments

This article describes the isolation of a new species of the enigmatic protist lineage rhodelphids. Rhodelphids occupy a very interesting position in the tree of eukaryotes, because they are basal to the clade that includes red algae, green algae, and glaucophytes. Thus, a better knowledge of this group would increase our understanding of primary endosymbiosis. Adding a new species from the soil is certainly very interesting and useful for future research.

In general, the article is well written. I have only one general comment: this paper is not about protist ecology, therefore, there is no use in writing (several times) that rhodelphids play an important role in soil ecosystems and their fertility, and that they can be used to control plant parasites. There has been no experimentation in this work that provides results in that sense... and it is not even clear whether rhodelphids (which are not very abundant in the environment) really play an important role. The work in interesting per se, there is no reason to find more arguments, the work is interesting enough to deserve publication!

Here are some additional comments
Lines 21-25: I would suggest removing the first part of the abstract. It's generic text, and the work deals with protist evolution, not ecology. I would rather replace them with a short statement about deep-branching eukaryotic phyla and their importance in reconstructing early eukaryotic evolution.

Lines 44-46: Here again, I would remove these sentences. I think that the work on rhodelphids is interesting per se and does not need to be justified by their role in agriculture.

Line 60: I would not compare protists with bacteria diversity, they evolve in a very different way...

Lines 62-73: It's nice to have a paragraph about the protist's ecological role in the introduction (not so much in the abstract), although I would shorten this a lot. This is not the aim of this article.

Line 74: I would suggest "Rhodelphids are one of the most recently discovered and poorly studied protist lineages."

Line 95-96: They probably play a role as predators in their ecosystem foodwebs, but showing that their role is really relevant requires other experimentation. I would remove this part of the sentence as well.

Line 209: through the apical invagination

Line 252:Thatt would make two hemispheres...

Lines 267-268: That can be an explanation, or maybe complex mastigonemes appeared independently several times. Or maybe the mastigonemes already existed before the Archaeplastida+Cryptista clade. Stramenopiles have complex, often tripartite mastigonemes. Could it be that this structure is ancestral in eukaryotes, making it a plesiomorphic trait? I would discuss this possibility.

Line 300: Archaeplastida

Line 304: Why is this extremely interesting? Maybe because such organisms should be anaerobic and thermophilic, and can therefore help us understand metabolism evolution in rhodelphids? What would be the implications? Please discuss that point.

Lines 309-310: Again, here, the "enormous impact" needs to be demonstrated... in another work!

Lines 329-352: This part is out of the scope of the paper. I would remove the whole paragraph.

Line 366: How do we know that? Earlier in this article authors mention the predation of bacteria. It should be made clear why these organisms need eukaryotes for food. If this were the case, how did the first Archaeplastids engulf a cyanobacterium?

Lines 376-377: For reasons detailed above, please remove

Reviewer 4 ·

Basic reporting

This is a description of just the third species of a VERY important protist group, evolutionarily speaking. The study also includes useful information about the distribution of the group across habitat types (and geography).

The fundamental of the work are solid, and basic conclusions sensible. I have two general points of criticism: 1) about the accuracy of the some reporting of the methods; 2) There is a lot of general discussion of soil protists/ecology and to a lesser extent the evolutionary relevance of Rhodelphids, that is not materially advanced by the current study – I think it is fine to talk about these things in this paper; but keep it succinct, and especially devote the space in the Discussion to points where something is changed or directly influenced by the Results of this study.

Experimental design

the design of the study is good; There are some shortcomings in the reporting of the methods, as described in general comments - see below. One figure's presentation needs improvement (Figure 2) - see below.

Validity of the findings

The fundamental validity of the findings is good.

Additional comments

Specific comments:

Figure 3 (whole-mount TEM) shows images of material that seems to be prepared with a metal-shadowing technique. The materials and methods instead describe a negative staining technique. This inconsistency lowers trust that the other details of the methods are completely correct. Please ensure ALL the materials and methods are correct for the work reported in this paper (e.g. look out for residual inaccuracies that come from using text for similar works as templates).

285-300: As written, this part of the Discussion is partly ambiguous, and on literal reading some of it seems to be scientifically indefensible. Please delete or heavily rewrite, noting (1) sequence identity (i) within a group or (ii) between a group and other groups, are very different things and it is crucial it is clear which you mean, (2) that the last common ancestors of groups can be separated hugely in time and biology from first exclusive ancestors of groups. (3) it is not obvious that rRNA divergence should correlate with other biological differences.

329-381: The last paragraph of the ‘main’ discussion, is mostly (lines 331-348) ‘introduction’ material on soil ecology/microbiology that is not affected directly by the results of the manuscript and should be greatly abridged at least. Then the ‘conclusions’ section includes some probably unnecessary summary of the morphology of R. edaphicus, followed by discussion/speculation on the ecological relevance of rhodelphids that includes parts that are not really covered elsewhere in the discussion, and then more general material on soil ecology that arguably does not have much directly to do with Rhodelphis. The authors might consider not having a section called “conclusions” at all, and condensing these two paragraphs into one polished paragraph that includes the best material on ecological (including soil ecology) considerations, especially those about Rhodelphis directly and/or where Rhodelphis could be most interesting.

Figure 2: Display of the images is too dark overall, with D being especially problematic.


Minor comments:

*“Predatory” is sometimes used in the protist literature as shorthand for “Eukaryotophic”, whereas here I think it is used to cover bacterivory as well. This is fine, but define what You mean the first time you use ‘predatory’ in the main text.

25-27: Suggest “One of the most enigmatic and poorly studied predatory protist groups is Rhodelphidia. These are heterotrophic flagellates, yet belong to the primarily photosynthetic supergroup Archaeplastida and are related to red algae in particular.”

29: phagocytoses

54-56: This reads oddly; Protists are the eukaryotes other than fungi, animals and plants. So not one group, and it is unremarkable to be ‘one of the most important’ if there are only 4 options!

57: delete ‘recent’. These references are all 8-12 years old

70: delete “with their broad prey range”; redundant in sentence.

74-75: Suggest “One of the most unusual and poorly studied predatory protist groups is rhodelphids (Rhodelphidia), which are unicellular, nonphotosynthetic biflagellates that live in aquatic environments”

76: “Strikingly, phylogenomic analyses revealed that the predatory..”

80: “..along with a relict nonphotosynthetic primary plastid that lacks a genome.” ?

83: “…suggesting that a mixotrophic nutritional mode, including phagotrophy, persisted well into the evolutionary history of the supergroup (Gawryluk et al., 2019).”

94-96: This does not follow logically, as would be implied by “thus”, and seems unsupported. What if they are always rare in nature?. delete, or substantiate and reword.

96-97: “..it is likely that this distinct phylum of eukaryotes includes significant…”

123: “Strains are currently stored in the collection of the AquaBioSafe Laboratory..”

137: “the filters” is an error? (judging from rest of protocol).

145: should “following peak” be “near peak” ?

151 “EukA and EukB”

177-185: Add description of how site selection for analysis was done after alignment, how many sites were retained for analysis, how many sequences are in the alignment (noting that this is not obvious from the figures), and how convergence of the Bayesian analysis was verified.

180-181: delete “to calculate posterior probability”

197: delete “differently”

200: “..flagellar pockets, which are separated..“

201-202: reads confusingly and ‘a complex of’ is too vague. Suggest “….but are equipped with thin bi- or tripartite mastigonemes that are arranged… [add short description of how arranged]”

203: “10.7-20.4” (also replace commas with dots in numbers later in line)

210 (and elsewhere?): “posterior end” would be clearer than “distal end” and is the natural contrast to the “anterior end”, used earlier.

212-213 (and 216?): This text seems to conflict with the videos and the previous sentence. It looks to me that phagocytosis is at the posterior end of the groove specifically. Reword slightly?

216: Rhodelphis [note: check for typos throughout text – I likely missed some]

220-221: These grooves are subtle and given the fixation quality could be more-or-less artefacts. Delete? If kept, make sure that you specify that they are only seen in SEM (unless you have LM showing them, but then label it), and give more details of number, and a caveat about maybe artefactual;

229: “92.7% identical”

230: “Together with the other…”

231: say which general habitat(s) these environmental sequences come from (soil / marine / freshwater / whatever)

232: specify that this % bootstrap support on first use.

262: reference(s)

262: “…we were unable to determine which type is most similar to those of R. edaphicus”

268: delete “already”?

269: delete “distinctive”: later “distinguishes” makes redundant.

273: reference. Also, Video S2 looked like there might well be pseudopodial activity.

276-279 and 280-281; These details should (also) be in the results.

300: Archaeplastida

329-381: as discussed above, I recommend this section be substantially rewritten.

349: “..soil protists like R. edaphicus...,,”

365-367: (i) If R. edaphicus cannot survive on bacteria alone, please make this very clear in the results (and the evidence as to why you know that). (ii) Now you have looked for Rhodelphis sequences in diverse environmental datasets, you do have some actual data related to their abundance, however imperfect. What do these data say?

395: Delete this line: it is confusing because ‘type figure’ is not an official thing – this figure is no more ‘type’ than ~all the other figures in the paper.

397: Give geographic coordinates here too, if possible.

399-401: The Hapantotype will include prey cells probably (and prokaryotes). I would flag these and indicate that they are excluded. Also it would be preferable that this were accessed in a collection in an institution with indefinite life and an accessibility policy, not in a person’s lab wherein they will retire. But maybe this is already the case.

408: I guess this is the LSID for the ‘work’ (required) and not for the new taxon (optional to provide?). To avoid confusion, please make clear in the text that this is the LSID for the ‘work’, and consider also including (as a separate explained entry) the LSID for the new species (because why not?)/

524: “Protozoa” not in italics

Figure 6 Legend currently says ‘red dots’; some of these dots are orange. Bring to mutual consistency somehow.

---

## Round 0.2 · accepted · Accept

· Academic Editor

Accept

Thanks for addressing all comments!

Reviewer 1 ·

Basic reporting

The manuscript was revised in accordance with my comments. I do not have any further concerns.

Experimental design

-

Validity of the findings

-

Reviewer 3 ·

Basic reporting

The authors responded satisfactorily to all queries by other reviewers and me.

Experimental design

-

Validity of the findings

-